# A Multi-Scale Feature Pyramid Network for Detection and Instance Segmentation of Marine Ships in SAR Images

Zequn Sun [1,†], Chunning Meng [2,†], Jierong Cheng [1], Zhiqing Zhang [1,3,*] and Shengjiang Chang [1]

1 Institute of Modern Optics, Nankai University, Tianjin 300350, China
2 China Coast Guard Academy, Ningbo 315801, China
3 State Key Laboratory of Applied Optics, Changchun Institute of Optics, Fine Mechanics and Physics, Chinese Academy of Sciences, Changchun 130033, China
* Correspondence: zhiqing.andy_zhang@nankai.edu.cn
† These authors contributed equally to this work.

**Abstract:** In the remote sensing field, synthetic aperture radar (SAR) is a type of active microwave imaging sensor working in all-weather and all-day conditions, providing high-resolution SAR images of objects such as marine ships. Detection and instance segmentation of marine ships in SAR images has become an important question in remote sensing, but current deep learning models cannot accurately quantify marine ships because of the multi-scale property of marine ships in SAR images. In this paper, we propose a multi-scale feature pyramid network (MS-FPN) to achieve the simultaneous detection and instance segmentation of marine ships in SAR images. The proposed MS-FPN model uses a pyramid structure, and it is mainly composed of two proposed modules, namely the atrous convolutional pyramid (ACP) module and the multi-scale attention mechanism (MSAM) module. The ACP module is designed to extract both the shallow and deep feature maps, and these multi-scale feature maps are crucial for the description of multi-scale marine ships, especially the small ones. The MSAM module is designed to adaptively learn and select important feature maps obtained from different scales, leading to improved detection and segmentation accuracy. Quantitative comparison of the proposed MS-FPN model with several classical and recently developed deep learning models, using the high-resolution SAR images dataset (HRSID) that contains multi-scale marine ship SAR images, demonstrated the superior performance of MS-FPN over other models.

**Keywords:** multi-scale feature pyramid network (MS-FPN); multi-scale feature maps; ship detection; ship segmentation; synthetic aperture radar (SAR)

## 1. Introduction

Synthetic aperture radar (SAR) is an important form of radar that is widely used in the remote sensing field to capture two-dimensional images or to create three-dimensional reconstructions of objects such as marine ships and natural landscapes [1]. As an active imaging sensor type using microwaves, SAR imaging is superior to traditional passive imaging sensors such as infrared and optical sensors from many aspects because it is less affected by environmental factors such as weather, visible light, and clouds. In marine affair management, SAR imaging plays an important role because it has the capability of detecting hidden objects and working in all-day and all-weather conditions [2–4]. With the rapid development of spaceborne and airborne SAR, for example TerraSAR-X and RADARSAT-2, SAR imaging has been routinely used for marine monitoring, fishery management, marine traffic control, marine emergency rescue, and so on [5,6], all of which unavoidably must have marine ship involved. Therefore, SAR data analysis relevant to marine ships, especially marine ship detection and segmentation, has become an important research direction in the remote sensing field, which is under active investigation in the recent years [7,8].

The detection and the segmentation of marine ships from SAR images, by detecting only the location and by outlining the accurate shape, respectively, is a challenging task. The challenge mainly arises from the multi-scale property of the marine ships presented in a SAR dataset and the accompanying complex backgrounds. As illustrated in Figure 1, marine ships in a SAR dataset can be very small and have variant sizes, which is due to their different categories and sizes, as well as the intrinsic imaging parameters of SAR imaging such as resolution and incident angle [9–11]. The number of marine ships present in two SAR images can also be very different. Moreover, the backgrounds in some SAR images are sometimes very complex because of the presence of inshore buildings, making the analysis of inshore marine ships much more challenging than offshore marine ships. Furthermore, we also noticed that the marine ships with a small size are the majority of some well-known SAR datasets, as illustrated in Figure 2 which is plotted from the high-resolution SAR images dataset (HRSID) [12]; therefore, the performance of detecting and segmenting small marine ships is crucial.

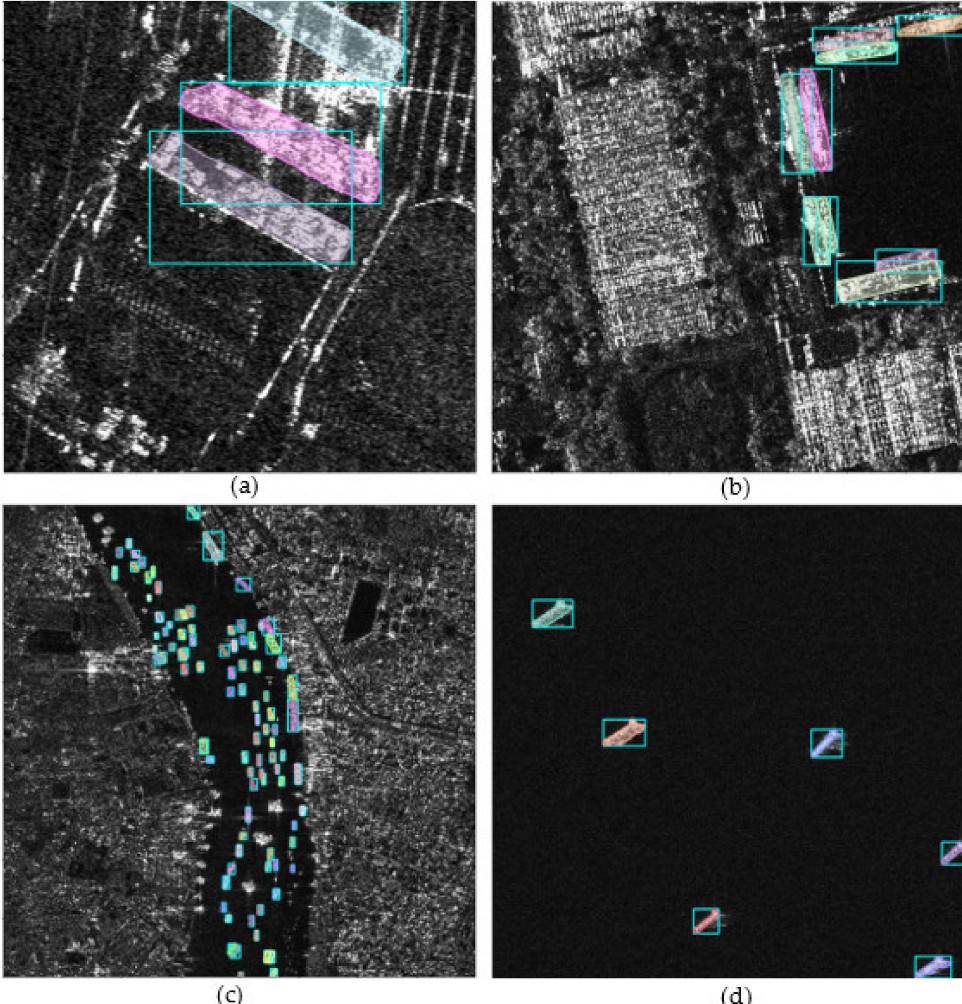

**Figure 1.** Representative examples of SAR images, showing marine ships of different scales in a complex background. The locations and segmentations of marine ships are outlined by colored rectangles and closed curves. (**a**) A SAR image representing large ships located nearby berthing buildings. (**b**) A SAR image representing middle-scale ships close to berthing buildings. (**c**) A SAR image representing small ships close to berthing buildings. (**d**) A SAR image representing the small ships in the offshore background.

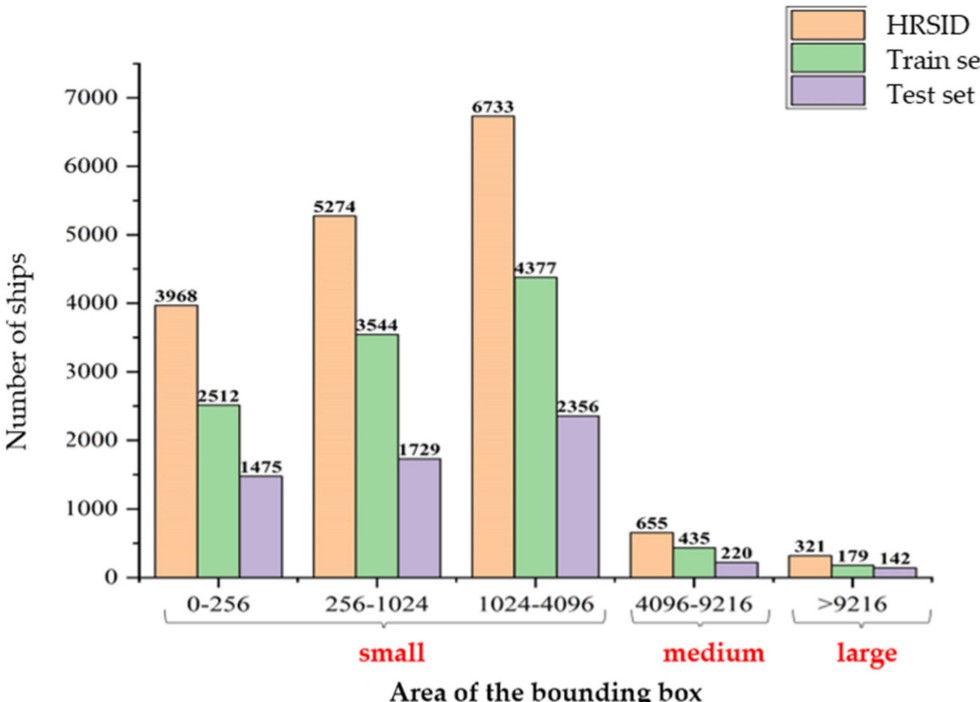

**Figure 2.** Statistics of the area of marine ships in the HRSRD dataset, showing that ships of a small scale are the majority of the dataset, and showing the importance of detecting and segmenting small marine ships.

Traditional methods for analysis of SAR images are mostly detection methods based on thresholding [13–15]. The constant false alarm rate (CFAR) detection algorithm and its successors are the most widely studied and applied thresholding methods [7,14,16,17]. Specifically, Gao et al. proposed a CFAR algorithm, using a decomposition strategy by incorporating a generalized gamma distribution into the CFAR detector, to improve the signal-to-noise ratio of SAR images, leading to improved detection accuracy [14]. Wang et al. presented an intensity-spatial domain CFAR algorithm which could make use of the intensity and spatial information of SAR images to enhance image contrast and the detection performance [18]. However, these traditional methods are usually designed in a very complex and cumbersome way that involves manual steps; thus, they are less applicable to SAR images with multi-scale marine ships appearing in a complex background. Moreover, traditional algorithms may only be applied on a limited number of images, and the feature analysis using this small dataset cannot reflect the multi-scale characteristics of marine ships in SAR images, leading to poor detection performance of multi-scale ships within complex backgrounds.

Convolutional neural network (CNN) also has been emerging as a potential solution for solving this complex multi-scale detection problem for SAR images, because of its capabilities of representing and learning features in multi-scale levels. Several CNN-based marine ship detection methods have shown promising performance on SAR images [19,20]. Rostami et al. presented a transferring knowledge framework to train a deep neural network for classifying SAR images by eliminating the need for a huge labeled dataset, achieving efficient and competitive results [21]. Several works presented real-time ship detection models to meet SAR image processing at mobile terminals [22,23]. In 2017, Lin et al. proposed the feature pyramid network (FPN) [24], which has become a standard solution for the detection of multi-scale marine ships in SAR images. FPN can detect ships having different sizes, imaged at different resolutions, using some reasonable semantic features extracted from its backbone networks, leading to promising performance, and thus has received a wide range of attentions [6,25], but FPN cannot adaptively learn and select salient feature scales for multi-scale ship detection in SAR images, leading to some false

and missed detections. To overcome this issue, the attention mechanism has been applied to analyze and amplify the salient ships, by ignoring irrelevant features [26–28]. Cui et al. proposed a variant FPN model that densely connects convolutional block attention module (CBAM) to individual concatenated feature maps from top to bottom of the pyramid network [29]. This model can extract high-resolution fused feature maps with richer semantic information that is useful for the detection of multi-scale marine ships. A squeeze and excitation module (SENet) was added to the top layer of FPN by Lin et al. in 2019, which was shown to outperform the primary FPN model [30]. In order to detect marine ships with varied sizes from complex backgrounds, Zhao et al. designed an attention-receptive pyramid network, via combining receptive field blocks and CBAM in the attention-receptive block to build a top-down fine-grained feature pyramid [31]. However, these attention methods only perform weight recalibration for input feature maps at a single scale and the single-scale attention mechanism cannot effectively capture all the important feature maps having different scales, while multi-scale is an intrinsic property of marine ships in SAR images. CNN-based models have also been invented for ship detection in other types of remote sensing data such as optical remote sensing data [32–36], but optical remote sensing data are very different from SAR data; thus, the invented models might not be applicable for SAR data.

To improve the detection of multi-scale marine ships in SAR images, some pioneer works have employed instance segmentation to assist the ship detection. Su et al. presented a high-resolution feature extraction network that was originally designed for instance segmentation of general remote sensing images [25], while SAR images were merely used as the testing data, achieving relatively good performance on segmenting marine ships. Nonetheless, this model was not specifically designed for SAR image analysis; it did not consider the multi-scale characteristics of marine ships, being inapplicable of accurately segmented multi-scale marine ships from a complex background. Wei et al. showed a high-resolution SAR image dataset (HRSID) for detection and instance segmentation of marine ships, but there were no accompanying instance segmentation methods [12]. To improving the segmentation performance of multi-scale ships, the attention mechanism was also applied to SAR image segmentation. Gao et al. introduced the CBAM module into the feature fusion process of FPN to extract salient features of different scales, thereby enhancing the ability of feature representation and lowering the interference of irrelevant information from a complex background [37]. Although CBAM can improve the segmentation performance by applying different weights to the input features [38], CBAM is a single-scale attention mechanism as just mentioned, which cannot efficient address the multi-scale ship segmentation problem of SAR images.

More recently, simultaneous instance segmentation and detection have gradually attracted the interest of scientists in the computer vision field [39–42]. Some pioneer works have attempted to address the problem of simultaneous detection and segmentation of multi-scale marine ship in SAR images, but there is still ample room for further addressing this problem. Specifically, Ke et al. proposed a global context boundary-aware network for SAR ship instance segmentation [43], and Zhang et al. proposed a hybrid task cascade plus model for SAR ship instance segmentation [44], but these single-scale attention mechanisms also cannot effectively capture all the important feature maps having different scales. Zhang et al. proposed a model that was dedicated to ship detection and instance segmentation of SAR images, by embedding a contextual SENet module into FPN to capture the prominent contextual information of the background from different levels [45]. However, two aspects limit the performance of this model and other existing models for simultaneous detection and instance segmentation of multi-scale marine ships. First, SENet is a single-scale attention mechanism as aforementioned. Second, these FPN-based models mainly use the deep semantic information, while the shallow high-resolution feature maps contain richer details that are crucial for detecting small ships in complex backgrounds. Therefore, there is an urgent demand to design an efficient model that: (1) can efficiently extract both the shallow and deep feature information which can be combined to achieve

the multi-scale representation of marine ships, (2) possesses a novel multi-scale attention mechanism to adaptively learn and select important features which will greatly facilitate both the detection and segmentation of multi-scale marine ships for SAR images.

In this paper, we present a multi-scale feature pyramid network (MS-FPN) for simultaneous detection and segmentation of marine ships in SAR images. MS-FPN mainly consists of two parts, the atrous convolutional pyramid (ACP) module that extracts shallow multi-scale feature maps for detection and instance segmentation of small objects from a complex background, and the multi-scale attention mechanism (MSAM) module that adaptively selects important multi-scale feature maps for detection and segmentation of multi-scale marine ships.

## 2. Methods

We present the details of the proposed MS-FPN model in this section. We will first analyze the architecture of the original FPN model and its drawbacks, which has inspired the emergence of MS-FPN. Then, we will describe in detail the overall architecture of the MS-FPN model, and the design of the ACP module and the MSAM module, respectively.

### 2.1. The Architecture of the Proposed MS-FPN Model

The proposed MS-FPN model is a variant model of FPN. The FPN model is designed in a top-down manner by using lateral connections to acquire a fine-grained feature pyramid [24]. Due to its ability to fuse multi-level features from the backbone network, the architectures of FPN have been widely used in many computer vision applications including marine ship detection and segmentation of SAR images. Nonetheless, one vital weakness of FPN and the existing FPN-based models is that the deep semantic features are more thoroughly used than the shallow features, making the detection of small ships inaccurate because the features of small objects are already smeared out in deep layers by the pooling operations. As shown in Figure 3a, compared with the abstract features extracted in deep layers, the features extracted by the filters in shallow layers contain more specific feature information, such as edges, textures, and prototypes, which is more useful for detecting and segmenting small ships. Moreover, as illustrated in Figure 3b, small ships have more pixels in shallow layer feature maps than in deep layers, meaning that more features of small ships can be used for analysis. Therefore, it is of great importance to combine both the shallow high-resolution feature maps with deep low-resolution feature maps for detecting and segmenting both the small and large ships in SAR images.

Another issue of the FPN-based models is that it is unclear which feature maps from which layers are more useful for detecting and segmenting multi-scale marine ships, after the extraction of multi-scale feature information. The single-scale attention mechanism models such as SENet and CBAM cannot distinguish feature maps obtained from different scales, while multi-scale feature maps are important for the detection and segmentation of multi-scale ships in SAR images (Figure 4). The multi-scale feature maps can be used to improve the performance of ship quantification from two aspects: (1) distinguishing a ship from the background by using a larger receptive field, and (2) perceiving information of small ships by using a smaller receptive field. For example, the ocean is a useful background containing contexture information that can be used to determine whether a smaller scale object placing over the ocean is a boat or a building. Thus, in order to learn and select the most important features from the multi-scale feature maps, designing a multi-scale attention mechanism is of particular importance for the detection and segmentation of marine ships in SAR images.

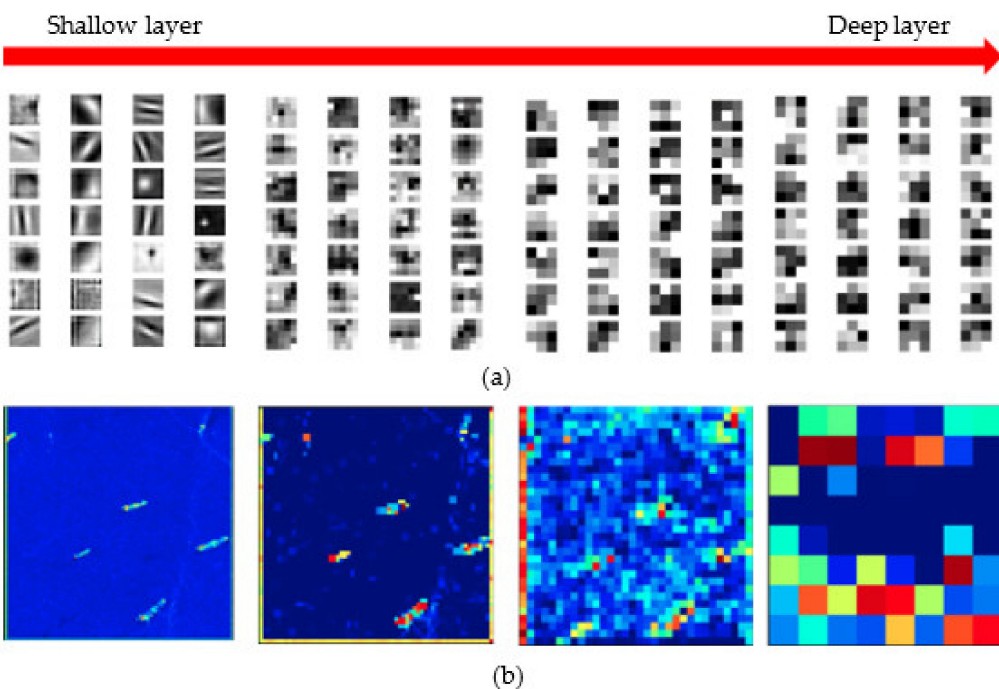

(a)

(b)

**Figure 3.** The visualization of feature maps of SAR images. (**a**) The visualization of feature maps extracted by filters from the backbone network of FPN. The shallow layers include more detailed information such as edges, circles, spots, and so on, while the information shown in the deep layers is more abstract. (**b**) Visualization of the feature maps of small ships from shallow layers to deep layers. The number of pixels of small ships decreases with the sizes of the feature maps.

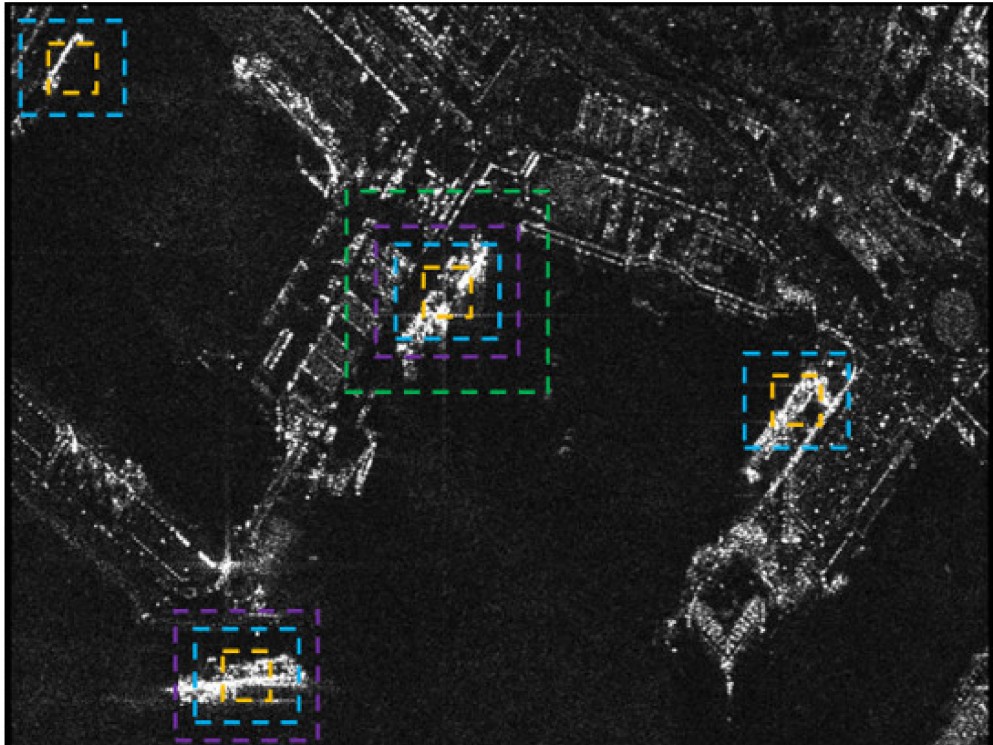

**Figure 4.** Multi-scale representations are critical for the detection and segmentation of marine ships in SAR images. The dashed boxes of different colors represent the extraction of features at different scales. These multi-scale representations can capture ship size, shape, and the background contexture, which are important information for ship detection and segmentation.

Here, aiming to combine the shallow and deep feature maps and to adaptively select important feature maps from the multi-scale feature maps, we design a multi-scale feature pyramid network (MS-FPN) specially for the accurate detection and segmentation of multi-scale ships from complex backgrounds in SAR images. The overall architecture of MS-FPN is shown in Figure 5, consisting of the atrous convolutional pyramid (ACP) module and the multi-scale attention mechanism (MSAM) module. We use ACP to extract the feature information of each stage between stage one and stage four, and we set the input channels of each stage of ACP to (256, 512, 1024, 2048), which are used for each stage of the backbone network. The number of channels is aligned, and the number of output channels is set to 256 for input alignment of top-down FPN. Then, we add the extracted feature information into top-down FPN for feature fusion, and we finally set the input and output channels of the MSAM module to 256 for consistency with the output of the top-down FPN. For example, we use the ACP module to extract shallow feature maps of stage two, after up-sampling the layer of P2 in the feature pyramid. Additive fusion is performed at the P1 scale, followed by an MSAM module for final prediction.

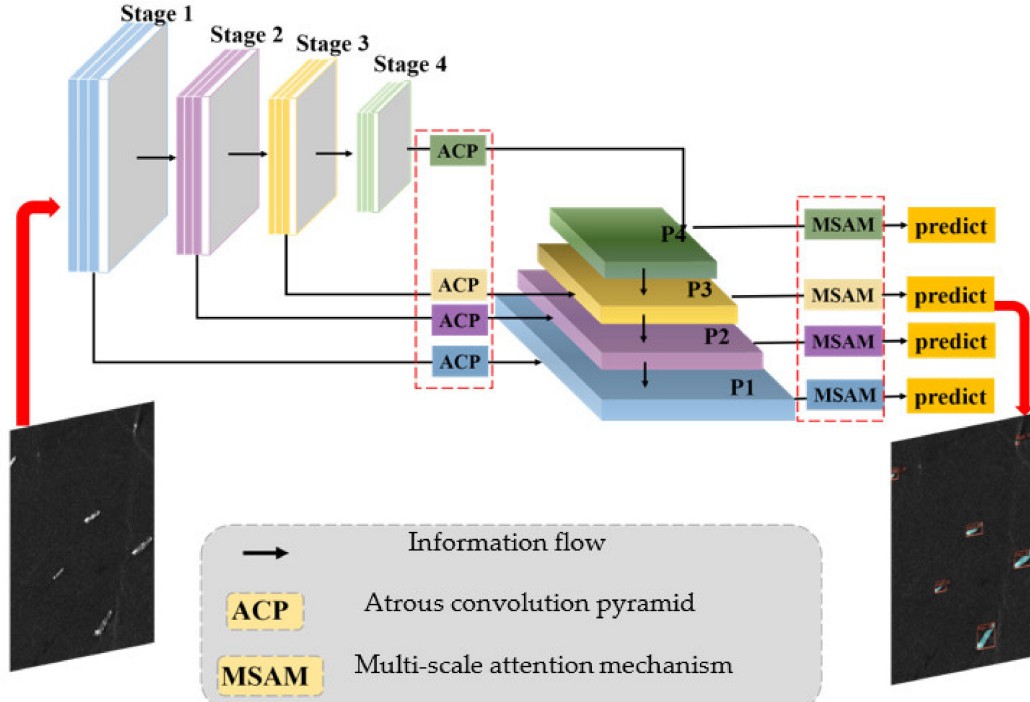

**Figure 5.** The flow chart of the proposed method. The core idea of this method is to embed ACP and MSAM into FPN, where ACP represents our proposed atrous convolution pyramid, and MSAM represents our proposed multi-scale attention mechanism module. The red dotted box represents our improvement.

Overall, the MS-FPN model has three advantages over the FPN model. First, using the ACP module with the capability of extracting multi-scale features, we are able to extract as much feature information as possible from the shallow high-resolution feature maps to improve the detection and segmentation of small ships. Second, with the MSAM module, MS-FPN can assign greater weights to certain feature maps that are more significant for ship detection and segmentation; thus, it is beneficial to eliminate the interference of the complex background in the SAR image. Third, both modules are based on multi-scale modules that can better deal with geometric problems such as scale transformation, which will further improve multi-scale ship detection and segmentation.

### 2.2. The Atrous Convolution Pyramid Module

Since FPN only uses $1 \times 1$ convolution to extract the shallow feature maps, the visual field of view is not large enough and the shallow features are not used sufficiently. Here, the ACP module is employed to obtain the complex contextual information from the last layer of each stage for detection of small ships, which is achieved by using convolutional kernels of different dilation rates. The core idea of ACP is to use parallel branches of multi-scale receptive fields, represented by different dilation rates, to extract multi-scale contextual information. As shown in Figure 6a, ACP mainly has three parallel $3 \times 3$ convolutional layers with dilation rates of 1, 2, and 4, respectively, meaning different receptive fields, which add multi-scale contextual information for the detection of small objects. This process of extracting feature maps is illustrated in Figure 6b. We reason that the use of varied dilation rates, according to the characteristics of small ships in SAR images, will reduce the background noise that may be introduced by using only one large dilation rate. The three parallel layers can be expressed as follows:

$$y = Conv_{3 \times 3, d=1}(x) \oplus Conv_{3 \times 3, d=2}(x) \oplus Conv_{3 \times 3, d=4}(x) \tag{1}$$

$$\text{and } out = Conv_{1 \times 1, d=1}(y), \tag{2}$$

where $d$ represents the dilation rate, and $\oplus$ is the operation of concatenation. $x$ denotes the input image, and *out* represents the output image. Due to the use of three parallel $3 \times 3$ convolutional layers with different dilation rates, ACP can increase the receptive field of the proposed model, which can significantly improve the performance in detecting small marine ships. Then, we merge these features together for feature refinement by element-wise addition of the three parallel layers. The final $1 \times 1$ convolutional layer performs channel dimensionality reduction to keep the output having the same channel dimension as the input of the next step.

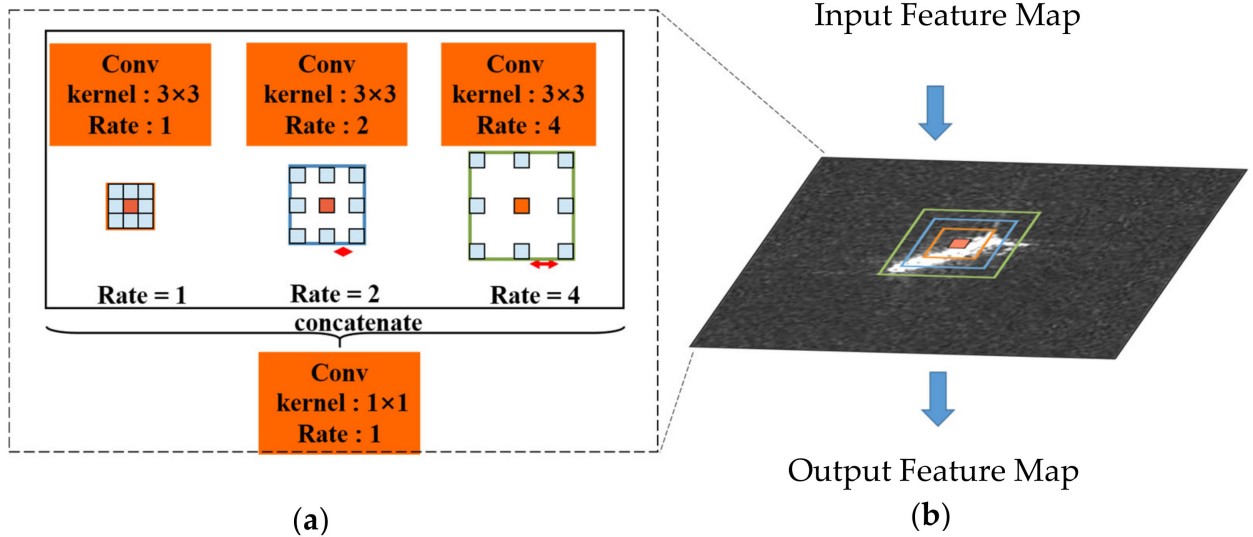

**Figure 6.** The schematic design of the ACP module. (**a**) The network structure of a specific ACP. (**b**) A schematic diagram representing the process of extracting feature maps at different scales. Different dilation rates mean different receptive fields.

Intuitively, the use of ACP has two merits: (1) the receptive field increases without sacrificing any detailed information, and (2) using three parallel channels can refine low-level feature representations and transfer contextual information from backbone to FPN, which is useful for the detection and segmentation of small ships. These merits of using ACP are illustrated in Figure 7, in which a bright background spot is falsely detected by FPN as a small ship (Figure 7a,b, the red circle), while the FPN model having an ACP module can exclude this false detection (Figure 7c). Using only FPN and deep low-resolution features

tends to regard noises similar to ships (possibly islands and reefs) as ships, resulting in false detections. ASPP is another multi-scale feature extraction module that is widely used in the field of computer vision [46], but having a large receptive field is opt to extract background noise features, which is not conducive to small ship detection in SAR images. The proposed ACP module is different from ASPP in two aspects: first, ACP uses convolution operation with lower void ratio, and second, ACP does not have the global pooling method. Therefore, ACP can reduce the false detection of noise in the complex background, resulting in better performance on extracting features of small ships.

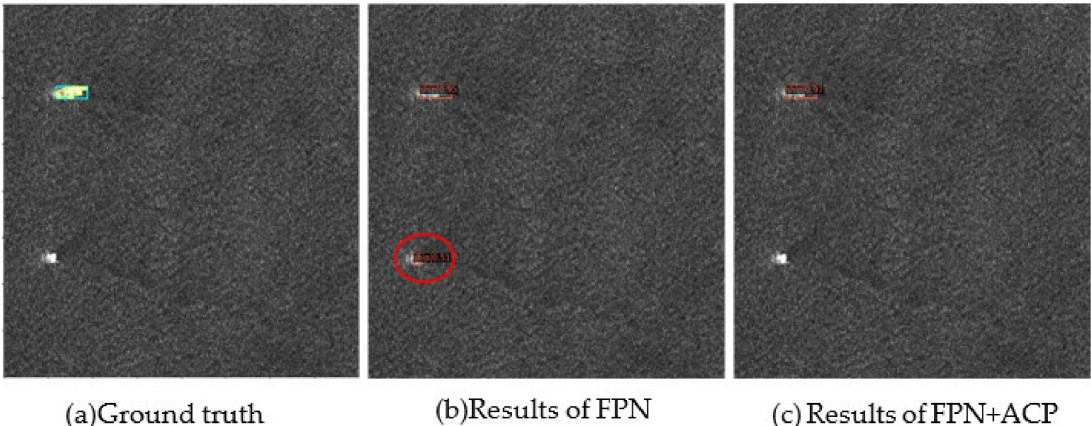

(a)Ground truth       (b)Results of FPN       (c) Results of FPN+ACP

**Figure 7.** Comparison of performance of FPN and FPN + ACP. (**a**) Ground truth. (**b**) The result of FPN. The red circle indicates a false detection by FPN. (**c**) The result of FPN + ACP, which is in agreement with the ground truth.

### 2.3. The Multi-Scale Attention Mechanism

In order to solve the segmentation problem of multi-scale ships in a complex background in SAR images and the issue of low efficiency in using multi-scale information by the current single attention mechanism models [32–35], here we propose MSAM, a multi-scale attention mechanism, to effectively utilize the multi-scale spatial information for improved detection and segmentation of multi-scale ships, by using the hierarchical residual-like structure and two full-connect layers.

The MASM is motivated by the following observation. The receptive field of a single $5 \times 5$ convolution is equivalent to the receptive field of two cascaded $3 \times 3$ convolutions (Figure 8). We then compare the number of parameters for the two types of convolutions, which is determined by,

$$params = C_{in} \times k^2 \times C_{out}, \tag{3}$$

where $C_{out}$ denotes the number of output channels, $C_{in}$ denotes the number of input channels, and $k$ denotes the size of the convolution kernel. For the same receptive field, the number of parameters for two $3 \times 3$ convolutions is much less than that for a $5 \times 5$ convolution; thus, we can use cascaded $3 \times 3$ convolutions instead of large convolution kernels to obtain the same receptive field. Similar to two concatenated $3 \times 3$ convolutions, the receptive field of three concatenated $3 \times 3$ convolutions is equivalent to the receptive field of a $7 \times 7$ convolution.

Motivated by this observation, the MSAM is built by different numbers of cascade $3 \times 3$ convolutions for multi-scale representations. The overall structure of MSAM is visualized in Figure 9. The construction of MSAM has three steps: (1) splitting and extracting multi-scale feature maps from the input image, (2) re-calibrating a weight according to the importance of a multi-scale feature map, and (3) concatenating multi-scale feature maps. More specifically.

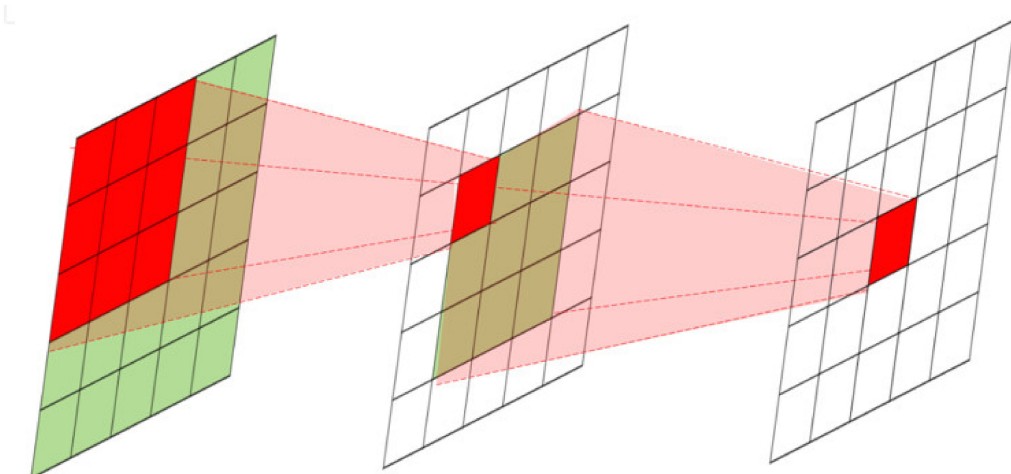

**Figure 8.** Schematic diagram of two cascaded $3 \times 3$ convolutions, whose receptive field is equivalent to a single $5 \times 5$ convolution.

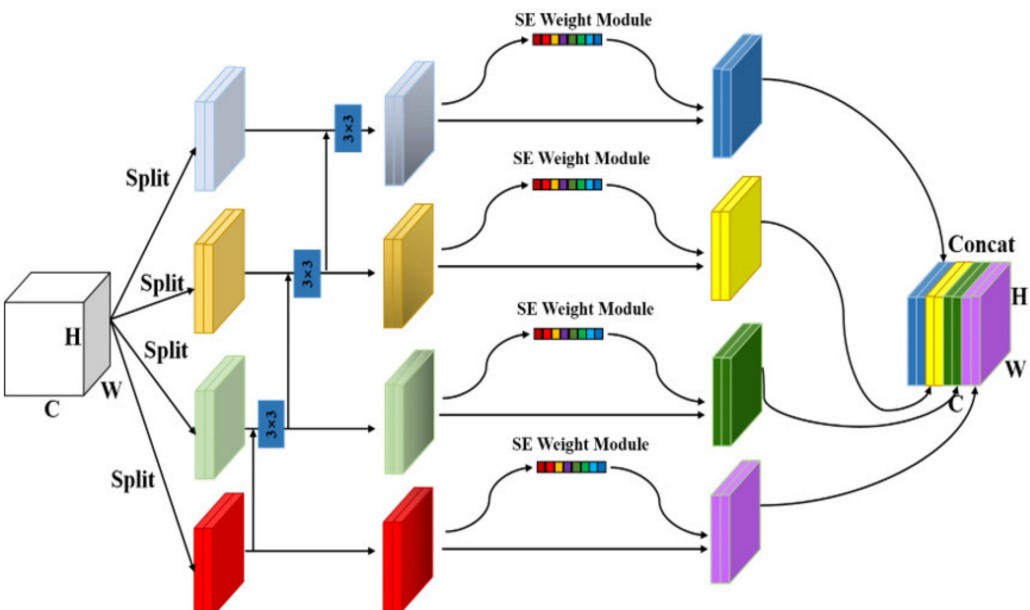

**Figure 9.** The structure of the proposed multi-scale attention mechanism. C, W, and H represent the value of channel, width, and height of the feature map, respectively. Split represents the split of the channels. $3 \times 3$ represents the size of a convolution kernel.

Step 1: multi-scale feature maps are extracted by using different numbers of cascades of small convolution groups. We split the original n-channel convolution into several small convolution groups with w channels (such that n = s × w). The small convolution groups are connected in a hierarchical residual-like style. The output of each small convolution group can be expressed as:

$$y_i = \begin{cases} x_i, i = 1; \\ K_i(x_i + y_{i-1}), 2 \leq i \leq s, \end{cases} \tag{4}$$

where $y_i$ denotes the output of the *i*-th group, $x_i$ denotes the input of the *i*-th group, and $K_i$ denotes the size of the convolution kernel of the *i*-th group. In each small convolution group, the input feature maps of the same receptive field finally generate equivalent multi-scale feature maps due to the different number of $3 \times 3$ convolution kernels.

Step 2: re-calibrating weights of each small convolution group of feature maps by two fully connected layers and pooling methods, which is used to obtain the channel correlation among multi-scale features. Average pooling and global pooling are applied to the feature maps of each small convolution group to obtain the global receptive field. The results can be presented as follows:

$$avg_i = \frac{1}{H \times W} \sum_{m=1}^{H} \sum_{n=1}^{W} y_i(m,n), \tag{5}$$

$$\max_i = \max_{(m,n) \in (H,W)} y_i(m,n), \tag{6}$$

where $y_i$ denotes the output of the $i$-th convolution group in step 1, $avg_i$ denotes the output of the average pooling, and $\max_i$ denotes the output of the max pooling. Then we perform a nonlinear transformation using two fully connected layers, and the activation function rectified linear unit (ReLU). After that, we sum up Equations (5) and (6) to generate the attention weights of the current small convolution group. The output of re-calibrating weights of feature maps can be expressed as follows:

$$att_i = w_{avgi} + w_{\max i}, \tag{7}$$

$$Y_i = y_i \times att_i, \tag{8}$$

where $Y_i$ denotes the output of the $i$-th convolution group after weight re-calibration.

Step 3: concatenating multi-scale feature maps. As shown in Figure 10, MASM is equivalent to the generated multi-scale attention feature map. The output can be expressed as follows:

$$out = concat([Y_0, Y_1, \ldots, Y_{s-1}]). \tag{9}$$

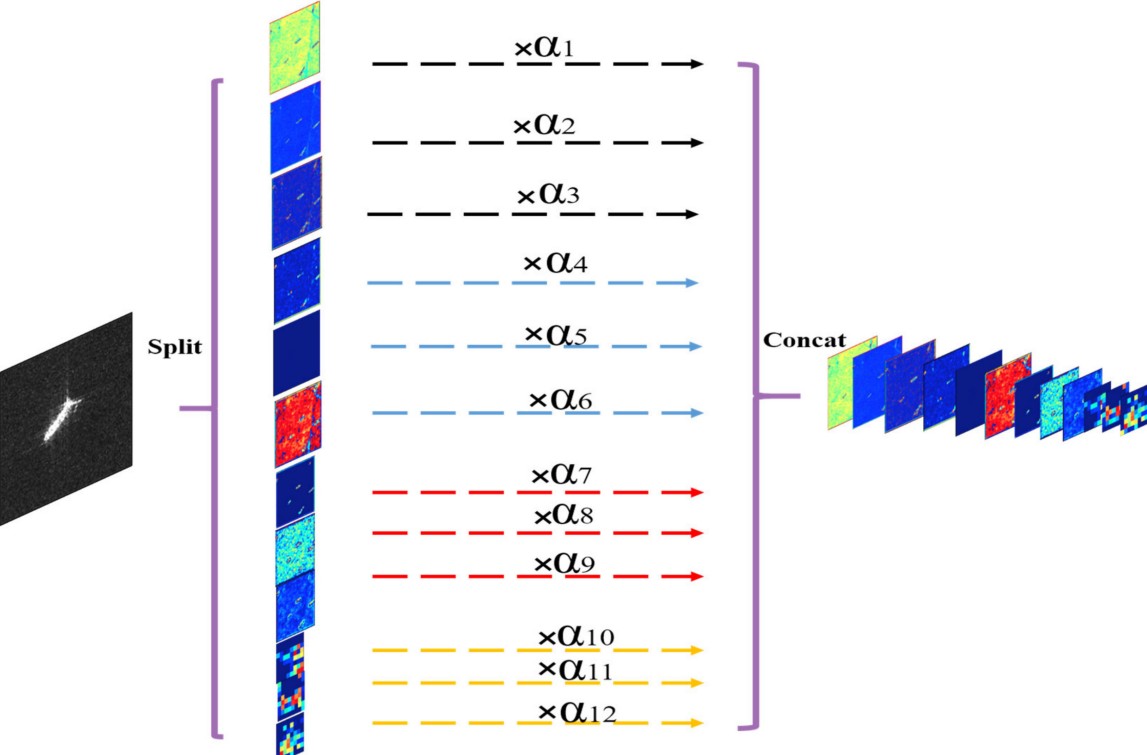

**Figure 10.** The diagram of MSAM, where $\alpha$ represents the weights applied to different feature maps.

In general, due to the use of MSAM, the extraction of feature maps at different scales can obtain more contextual information about multi-scale marine ships. Moreover, the employment of the re-calibrated weights makes the model learn and select the feature

maps that are relatively more useful for the ship detection and segmentation, and ignore the feature maps that hinder accurate detection and segmentation of marine ships. The merits of using MSAM is illustrated in Figure 11, in which the bright spots on the shore are falsely detected by FPN as small ships (Figure 11b, red circles), while the FPN model with a MASM module can exclude the false detections. The FPN model tends to regard noises similar to ships as ships.

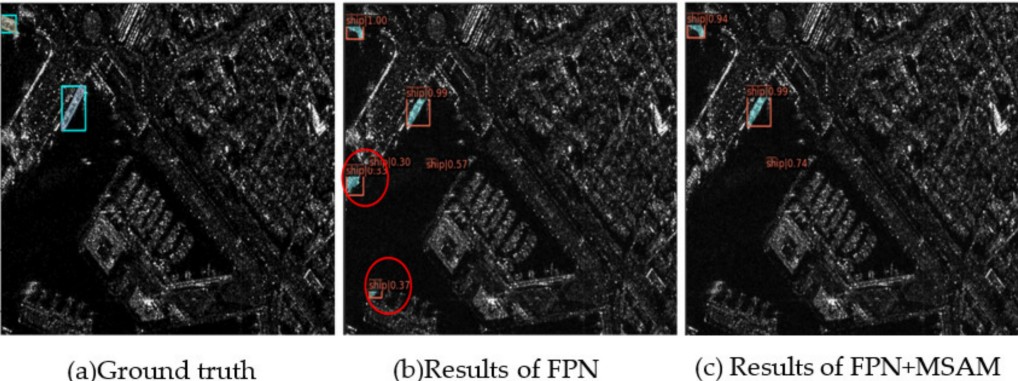

**Figure 11.** Comparison of performance of FPN and FPN + MSAM. (**a**) Ground truth. (**b**) The result of FPN. The red circles indicate the false detection by FPN. (**c**) The result of FPN + MSAM. In a complex background, FPN falsely recognizes many inshore buildings as ships, which can be avoided by the use of MSAM.

## 3. Experiments

### 3.1. DataSet and Settings

We used the public dataset HRSID [12], which had been accurately annotated, to train and test the performance of MS-FPN on ship detection and instance segmentation. We selected HRSID because it is the first SAR dataset of marine ships that supports instance segmentation. In this dataset, there are 136 panoramic SAR images, having resolutions between 1 m and 5 m, obtained from three different satellites, Sentinel-1B, TerraSAR-X, and TanDEM. These panoramic SAR images were cropped into $800 \times 800$ pixel images with an overlap rate of 25%. The dataset contains a total of 5604 cropped SAR images and 16,951 annotated ships.

Transfer learning is a technique to avoid training a neural network from scratch, and here we used the pre-trained ImageNet model as the start point for retraining [47]. The experimental environment was configured with Ubuntu 16.04.4, pytorch1.4.0, and 4 NVIDIA GeForce GTX 1080Ti GPUs. The classic Mask R-CNN was used as the base model of MS-FPN and other tested models [37].

### 3.2. Evaluation Criteria

We use average precision (AP) as an evaluation metric to measure the performance of a model. AP is defined as the integration of the precision over the recall rate:

$$\text{AP} = \int_0^1 P(R)\mathrm{d}R, \tag{10}$$

where $P$ represents precision and $R$ represents recall rate. Intersection of union (IoU) is defined as the overlap rate of the predicted bounding box and the ground truth bounding box. $AP$ is the primary challenge metric for the calculation of average IoU, which has ten thresholds distributed from 0.5 to 0.95 with an increment of 0.05. $AP_{50}$ and $AP_{75}$ represent the calculation under the IoU threshold of 0.5 and 0.75, respectively. In terms of the capabilities in multi-scale object detection, $AP_S$, $AP_M$, and $AP_L$ are used, denoting the precision of objects with small (area $< 32^2$ pixels), medium ($32^2 <$ area $< 64^2$ pixels), and

large (area $> 64^2$ pixels) size. In SAR images, the ships are multi-scale and most of the ships have a small size (Figure 2). Here, we keep $AP_S$ as the evaluation index.

### 3.3. Evaluation of MS-FPN

#### 3.3.1. The Detection and Segmentation Performance of MS-FPN

We compared the performance of our proposed MS-FPN model with several classic models, including FPN [24], FPN-carafe [48], HRFPN [49], and PAFPN [50], using the same environment configuration to ensure the fairness of the comparison.

The quantitative comparison results of detection and segmentation performance are summarized in Table 1. From these results, we found that that FPN has the worst performance in both detection and segmentation of multi-scale marine ships. The HRFPN of deep high-resolution representation learning, the PAFPN of path augmentation, and the CARAFE of content-aware reassembly algorithms show limited improvements in detection and segmentation of marine ships, and they cannot incorporate semantic information from different layers. The overall performance of MS-FPN outperforms other models, which is because MS-FPN can fully use the semantic information and detail information of all layers deeper than a certain layer, and can obtain high-resolution feature maps with more detailed semantic information.

**Table 1.** The detection and segmentation performance of FPN, FPN-CARAFE, HRFPN, PAFPN, and the proposed MS-FPN model. Note that the MS-FPN experiment was repeated five times and the values represent the average $\pm$ standard deviation.

| Mask R-CNN | Detection | | | | Segmentation | | | |
|---|---|---|---|---|---|---|---|---|
| | AP | $AP_{50}$ | $AP_{75}$ | $AP_S$ | AP | $AP_{50}$ | $AP_{75}$ | $AP_S$ |
| FPN [24] | 58.5 | 81.1 | 67.1 | 59.6 | 50.9 | 79.3 | 61.2 | 50.4 |
| FPN-CARAFE [48] | 58.6 (+0.1) | 80.7 | 67.1 | 59.7 | 51.0 (+0.1) | 78.8 | 62.1 | 50.6 |
| HRFPN [49] | 59.1 (+0.6) | 81.0 | 67.9 | 60.2 | 51.3 (+0.4) | 79.7 | 62.0 | 51.0 |
| PAFPN [50] | 59.3 (+0.8) | 81.4 | 68.2 | 60.3 | 51.4 (+0.5) | 79.4 | 62.0 | 51.0 |
| MS-FPN (ours) | 60.2 $\pm$ 0.1 (+1.7) | 82.3 $\pm$ 0.2 | 69.4 $\pm$ 0.4 | 61.3 $\pm$ 0.4 | 52.4 $\pm$ 0.1 (+1.5) | 80.3 $\pm$ 0.2 | 63.4 $\pm$ 0.5 | 52.0 $\pm$ 0.2 |

Visually, the proposed MS-FPN model also has the best performance on the SAR HRSID dataset in detecting and segmenting multi-scale marine ships from complex backgrounds, as shown in Figure 12. Compared with the ground truth (Figure 12a), FPN showed some false detections (red circle, Figure 12b), missed detections (yellow circle, Figure 12b), and poor segmentations (green circle, Figure 12b). FPN-CARAFE also showed a reasonable amount of missed detections (yellow circle, Figure 12c). The land clutter was falsely detected as a marine ship (red circle, Figure 12c). Similarly, false detections (red circle, Figure 12d,e), missed detections (yellow circle, Figure 12d,e), and imperfect segmentation (green circle, Figure 12d,e) were presented in the results of the HRFPN model and the PAFPN model. Compared to other models, the proposed MS-FPN model had less false detections and missed detections, and the segmentations were more accurate with a smoother outline (Figure 12f). Only MS-FPN can detect the locations of ships and segment the contours of ships correctly. Therefore, the proposed MS-FPN model has better performance in detecting and segmenting not only the large marine ships, but also small ships surrounded by a complex background. The improved performance of MS-FPN is due to the efficient extraction of shallow-layer feature maps by the ACP module and the fusion of these feature maps by the MSAM module, which can be combined with deep feature maps to achieve a significant improvement.

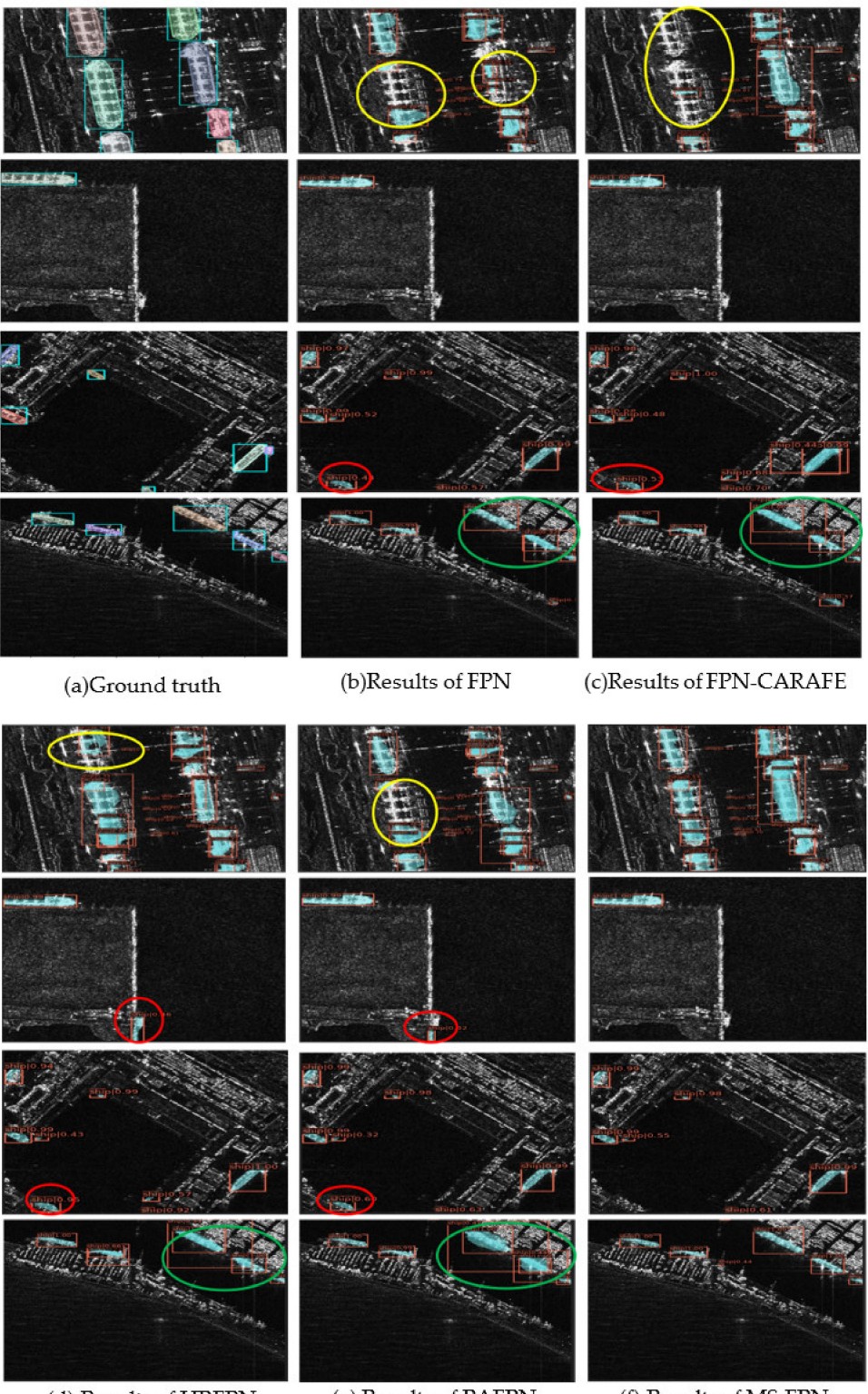

**Figure 12.** Comparison of FPN, FPN-CARAFE, HRFPN, and MS-FPN. (**a**) Ground truth. (**b–f**) Results of FPN, FPN-CARAFE, HRFPN, PAFPN, and MS-FPN, respectively. Red circles indicate false detections. Yellow circles indicate missed detections. Green circles indicate poor segmentation.

### 3.3.2. The Effect of Combining MSAM and ACP

To further evaluate which module (ACP or MSAM) contributes more to the improvement of the proposed MS-FPN model, we used Mask R-CNN as the basic model, and we compared performance of the models using none of ACP and MSAM, using only the ACP

module, using only the MSAM module, to the model using both ACP and MSAM modules. The quantitative comparison results in the detection and segmentation of marine ships are summarized in Table 2. Compared to the model using neither ACP or MSAM, the model with only the ACP or MSAM module showed improved performance in both detection and segmentation. Furthermore, the model using both the ACP and MSAM modules showed better performance than the other three mentioned above. This result shows that either ACP or MSAM alone can improve the performance of the model and the MSAM module contributes slightly more to the proposed MS-FPN model, while their combination can make full use of the shallow feature information and select the multi-scale feature maps that are relatively more beneficial for ship detection and segmentation, and therefore can further improve model performance. Nonetheless, here we also notice that the model with only MSAM can occasionally have slightly better performance than the model with both ACP and MSAM modules (see $AP_{50}$ in Table 2).

**Table 2.** The performance of models having MSAM or ACP modules.

| ACP | MSAM | Detection | | | | Segmentation | | | |
|---|---|---|---|---|---|---|---|---|---|
| | | AP | $AP_{50}$ | $AP_{75}$ | $AP_S$ | AP | $AP_{50}$ | $AP_{75}$ | $AP_S$ |
| × | × | 58.5 | 81.1 | 67.1 | 59.6 | 50.9 | 79.3 | 61.2 | 50.4 |
| √ | × | 59.7 | 81.6 | 68.6 | 60.6 | 51.9 | 79.8 | 62.4 | 51.4 |
| × | √ | 59.9 | 82.7 | 69.0 | 61.1 | 51.9 | 80.9 | 62.3 | 51.5 |
| √ | √ | 60.1 | 82.4 | 69.3 | 61.2 | 52.3 | 80.4 | 62.7 | 51.8 |

In order to visually compare the effectiveness of each module, we performed the ablation experiment on ACP, MSAM, and ACP&MSAM (Figure 13). Compared with the ground truth (Figure 13a), the model using only ACP showed both false detections (red circle, Figure 13b) and missed detections (yellow circle, Figure 13b). The model using only MSAM showed a reasonable amount of false detections (red circle, Figure 13c) and inaccurate segmentations (green circle, Figure 13c). As shown in Figure 13d, the model having both ACP and MSAM had less false detections and missed detections, and the segmentations were more accurate with a smoother outline. It is also important to note that the model with only the MSAM module can sometimes have better performance on detecting very tiny ships than the model with both the ACP and MSAM modules (see the second line of Figure 13), which indicates that the ACP module could reject the detection of some tiny ships by considering them as background noise (see Figure 7), when using the shallow high-resolution information to detect small ships. Taken together, these results prove that the combination of ACP and MSAM can better utilize the multi-scale feature maps of the network.

### 3.3.3. The Comparison of ACP with ASPP

We next moved to a detailed evaluation of the performance of each module (ACP and MSAM). We first compared the ACP module with the classical multi-scale feature extraction structure ASPP [46], and the results are summarized in Table 3. It can be seen that the AP values of ASPP are lower than those of ACP in both detection and segmentation. Meanwhile, the AP values of ACP and ASPP are higher than those of the base model Mask R-CNN. This comparison suggests that: (1) the shallow high-resolution features are useful for detecting and segmenting multi-scale ships in SAR images, and (2) ACP has better performance than ASPP. The underlying reason is that ASPP uses a larger dilated rate than ACP so that ASPP has a larger receptive field, which may lead to a higher level of background noise. It is also confirmed that although the increment of the receptive field can increase the performance of the base model, an excessively large receptive field can result in a negative impact on the performance of multi-scale ship detection and segmentation in SAR images.

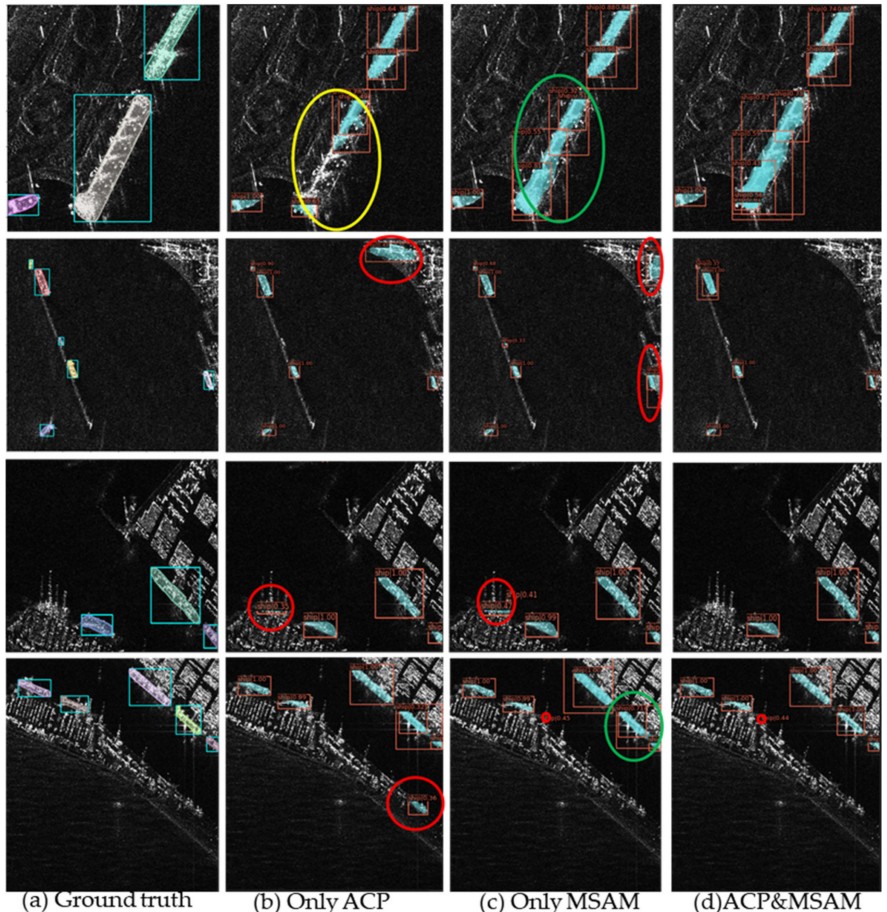

**Figure 13.** Visual results of models having only ACP or MSAM, and having both modules. (**a**) Ground truth. (**b**) The result of the model having only ACP. (**c**) The result of the model having only MSAM. (**d**) The result of the model having both modules. Red circles indicate false detections. Yellow circles indicate missed detections. Green circles indicate poor segmentation.

**Table 3.** The comparison of ACP and ASPP modules.

| Mask R-CNN | Detection | | | | Segmentation | | | |
|---|---|---|---|---|---|---|---|---|
| | AP | $AP_{50}$ | $AP_{75}$ | $AP_S$ | AP | $AP_{50}$ | $AP_{75}$ | $AP_S$ |
| Baseline | 58.5 | 81.1 | 67.1 | 59.6 | 50.9 | 79.3 | 61.2 | 50.4 |
| ASPP [46] | 59.2 | 81.5 | 67.5 | 60.0 | 51.7 | 79.7 | 62.5 | 51.2 |
| ACP | 59.7 | 81.6 | 68.6 | 60.6 | 51.9 | 79.8 | 62.4 | 51.4 |

### 3.3.4. Performance Comparison of MSAM

We then compared our proposed MSAM module with several classical attention mechanisms, including ECA-Net [28], SENet [38], CBAM [51], and CA [27]. The quantitative comparison results in detection and segmentation of marine ships are summarized in Table 4. Compared to the model without using an attention mechanism, the model using one of the four attention mechanisms (ECA-Net, SENet, CBAM, and CA) showed improved performance in both detection and segmentation. The model using MSAM showed better performance than the models using other attention mechanisms. This comparison leads to two conclusions. First, connecting an attention mechanism to a pyramid network is effective for detecting multi-scale ships in SAR images. Second, the multi-scale attention mechanism MSAM is more powerful in the detection and instance segmentation of multi-scale marine ships in SAR images, compared with the aforementioned single-scale attention mechanisms.

**Table 4.** Comparison of MSAM to classic attention mechanisms.

| Mask R-CNN | Detection | | | | Segmentation | | | |
|---|---|---|---|---|---|---|---|---|
| | AP | AP$_{50}$ | AP$_{75}$ | AP$_S$ | AP | AP$_{50}$ | AP$_{75}$ | AP$_S$ |
| Baseline | 58.5 | 81.1 | 67.1 | 59.6 | 50.9 | 79.3 | 61.2 | 50.4 |
| ECA-Net [28] | 59.1 | 81.3 | 67.5 | 60.1 | 51.4 | 79.4 | 62.3 | 51.0 |
| SENet [38] | 59.2 | 81.5 | 67.6 | 60.2 | 51.7 | 80.2 | 62.1 | 51.1 |
| CBAM [51] | 59.2 | 81.5 | 67.5 | 59.9. | 51.3 | 79.0 | 62.3 | 50.6 |
| CA [27] | 59.3 | 81.5 | 67.5 | 60.2 | 51.6 | 79.5 | 62.2 | 51.3 |
| MSAM (ours) | 59.9 | 82.7 | 69.0 | 61.1 | 51.9 | 80.9 | 62.3 | 51.5 |

### 3.3.5. The Effect of the Size of Receptive Field on Model Performance

To explore the impact of receptive field on the detection and segmentation performance of a model, we tested the performance with different scale factors that can obtain a receptive field of a specific size during convolution. As shown in Table 5, the size of the receptive field is important for model detection and segmentation. The scale of 4 achieved the best results for ship detection and segmentation, which are 0.4 AP and 0.2 AP, higher than the case of scale 2, with 0.6 AP and 0.2 AP, which are still higher than the baseline, respectively. The scale of 8 showed less improvement compared to the scale of 4. This decreased performance may be due to the introduction of additional noise by too large of a receptive field in the SAR images.

**Table 5.** The effect of scale variation on model performance.

| Mask R-CNN | Detection | | | | Segmentation | | | |
|---|---|---|---|---|---|---|---|---|
| | AP | AP$_{50}$ | AP$_{75}$ | AP$_S$ | AP | AP$_{50}$ | AP$_{75}$ | AP$_S$ |
| Baseline | 58.5 | 81.1 | 67.1 | 59.6 | 50.9 | 79.3 | 61.2 | 50.4 |
| S = 2 | 59.5 | 82.0 | 68.2 | 60.5 | 51.7 | 80.1 | 62.2 | 51.4 |
| S = 4 | 59.9 | 82.7 | 69.0 | 61.1 | 51.9 | 80.9 | 62.3 | 51.5 |
| S = 8 | 59.3 | 81.8 | 68.1 | 60.4 | 51.7 | 79.9 | 62.2 | 51.5 |

### 3.3.6. The Effect of Pooling Functions on Model Performance

Since pooling is an important part of MSAM, we used several different types of pooling functions to further study the impact of pooling on the performance of the model. As shown in Table 6, the pooling function is important for the model performance because models having a pooling function more or less showed performance improvement. Compared with the base model having neither an attention mechanism or a pooling function, the use of the max pooling function did not have an apparent improvement, but the use of the average pooling function showed an improvement of 1.4 AP and 0.9 AP in detection and segmentation, respectively. The mixed pooling of the max and average functions had the highest detection and segmentation improvement which is 1.4 AP and 1.0 AP, respectively. This comparison demonstrates that mixed pooling can gain more useful information on multi-scale ships than single pooling.

**Table 6.** The effect of pooling functions on model performance.

| Mask R-CNN | Detection | | | | Segmentation | | | |
|---|---|---|---|---|---|---|---|---|
| | AP | AP$_{50}$ | AP$_{75}$ | AP$_S$ | AP | AP$_{50}$ | AP$_{75}$ | AP$_S$ |
| Baseline | 58.5 | 81.1 | 67.1 | 59.6 | 50.9 | 79.3 | 61.2 | 50.4 |
| Max | 59.0 | 81.7 | 67.4 | 60.1 | 51.6 | 80.1 | 62.2 | 51.3 |
| Avg | 59.9 | 82.4 | 68.5 | 61.0 | 51.8 | 80.5 | 62.0 | 51.4 |
| Max and Avg | 59.9 | 82.7 | 69.0 | 61.1 | 51.9 | 80.9 | 62.3 | 51.5 |

### 3.3.7. Comparison with Other Advanced Models Using a Different Backbone and an Image Input Size

Since an image input of resizing the shorter side to 800 pixels is a widely used setting in the computer vision [52,53], the performance of MS-FPN was mostly tested using this image size, while some recent works on SAR ship detection of the HRSID dataset used an image size of $1000 \times 1000$ [12,54]. Moreover, these works used the ResNet-101 as the backbone, which is different from ResNet-50 used in this paper by now. In order to compare with these recent works, we have tested the performance of our proposed MS-FPN model, using the same setting, namely the input image size of $1000 \times 1000$ and the backbone of ResNet-101. The results are summarized in Table 7. In all, our MS-FPN model using the new setting (see the result of Mask R-CNN MS-FPN in Table 7) has superior to or comparable models than most models except Filtered Convolution, FL-CSE-ROIE, GCBANet, and HTC+. Then, we further explored the possibility of plugging the ACP and MSAM modules into the compared models such as Cascade Mask R-CNN and HTC, and we found that the generated new models all can improve the performance of the original models (see the results of Cascade Mask R-CNN-MS-FPN and HTC-MS-FPN in Table 7), resulting in better performance than the models Filtered Convolution and FL-CSE-ROIE. Finally, we also combined the HTC model with ACP and MSAM modules, yet using a different backbone ResNext-101-64xd (see the result in the last row of Table 7) [55], resulting in an improved result higher than all the models including GCBANet, except HTC+. It is unfortunate that the HTC+ model is not open source and we were not able to combine the HTC+ model with the proposed ACP and MSAM modules, and to test the generated model. Taken together, these findings demonstrated the performance of the proposed MS-FPN model as well as the ACP and MSAM modules.

**Table 7.** The comparison results with other advanced models.

| Method | Backbone | Detection | | | | Segmentation | | | |
|---|---|---|---|---|---|---|---|---|---|
| | | AP | AP$_{50}$ | AP$_{75}$ | AP$_S$ | AP | AP$_{50}$ | AP$_{75}$ | AP$_S$ |
| Mask R-CNN [53] | ResNet-101 | 65.1 | 87.7 | 75.5 | 66.1 | 54.8 | 85.7 | 65.2 | 54.3 |
| Mask Scoring R-CNN [56] | ResNet-101 | 65.2 | 87.6 | 75.4 | 66.5 | 54.9 | 85.1 | 65.9 | 54.5 |
| Cascade Mask R-CNN [52] | ResNet-101 | 65.1 | 85.4 | 74.4 | 66.0 | 52.8 | 83.4 | 62.9 | 52.2 |
| PANet [50] | ResNet-101 | 65.4 | 88.0 | 75.7 | 66.5 | 55.1 | 86.0 | 66.2 | 54.7 |
| YOLACT [57] | ResNet-101 | 47.9 | 74.4 | 53.3 | 51.7 | 39.6 | 71.1 | 41.9 | 39.5 |
| GroIE [58] | ResNet-101 | 65.4 | 87.8 | 75.5 | 66.5 | 55.4 | 85.8 | 66.9 | 54.9 |
| Filtered Convolution [59] | ResNet-101 | 68.6 | 89.2 | 77.6 | 67.4 | - | - | - | - |
| FL-CSE-ROIE [60] | ResNet-101 | 69.0 | 90.2 | 79.5 | 69.9 | 57.9 | 88.6 | 69.5 | 57.3 |
| GCBANet [43] | ResNet-101 | 69.4 | 89.8 | 79.2 | 70.4 | 57.3 | 88.6 | 68.9 | 57.0 |
| HTC [61] | ResNet-101 | 66.6 | 86.0 | 77.1 | 67.6 | 55.2 | 84.9 | 66.5 | 54.7 |
| HTC+ [44] | MRFEN | 71.5 | 92.3 | 82.5 | 72.6 | 59.1 | 90.3 | 71.0 | 58.7 |
| Mask R-CNN MS-FPN (ours) | ResNet-101 | 66.3 | 88.4 | 76.0 | 67.4 | 56.3 | 86.3 | 67.6 | 55.5 |
| Cascade Mask R-CNN-MS-FPN (ours) | ResNet-101 | 69.2 | 88.8 | 79.9 | 70.0 | 57.4 | 87.7 | 69.7 | 56.5 |
| HTC-MS-FPN (ours) | ResNet-101 | 69.2 | 89.2 | 79.4 | 69.9 | 57.6 | 87.4 | 69.3 | 56.7 |
| HTC-MS-FPN (ours) | ResNext-101-64xd | 70.1 | 89.4 | 80.9 | 70.7 | 58.5 | 88.2 | 71.6 | 57.7 |

## 4. Conclusions

The presence of multi-scale marine ships is the inner property of the SAR image dataset. The single-scale attention mechanism cannot effectively capture the multi-scale features of marine ships, resulting in low accuracy in detecting and segmenting small ships. In order to address this issue, we showed the MS-FPN model, a simultaneous detection and segmentation method for multi-scale ships, which can efficiently combine shallow high-resolution feature maps with deep low-resolution feature maps that are useful for detection and segmentation of multi-scale ships, especially the small ones. The MS-FPN

model consists of the ACP module and the MSAM module. The ACP module is able to extract shallow multi-scale feature maps and the MSAM module is able to adaptively learn and select the important multi-scale feature maps extracted by the ACP module. The performance of the proposed MS-FPN model was verified on the HRSID dataset that consists of high-resolution SAR images of marine ships, with comparison to several important CNN-based models (such as FPN, HRFPN, FPN-CARAFE, and PAFPN, etc.), several classic single attention models, as well as the classic multi-scale feature extraction model ASPP. The improved performance in the detection and segmentation of multi-scale ships from complex backgrounds demonstrated efficiency of the proposed MS-FPN model. In order to understand which module has a more significant contribution to the improvement of MS-FPN, we separately compared the performance of models having only the ACP module, the MSAM module, and then both of the ACP and MSAM modules, which shows that the MSAM module has a slightly higher contribution, while the combination can greatly enhancement the detection and segmentation accuracy.

Missing and false detection is a major problem in the currently existing models for SAR image detection and instance segmentation. Through the use of multi-scale feature information and multi-scale attention mechanism, the proposed MS-FPN model has significantly reduced the number of missing and false detections. Meanwhile, we also noticed that some very tiny ships are not detected by the MS-FPN model (see the second line of Figure 13). The ablation experiment showed that the model with only the MSAM module could detect those tiny ships, but the MS-FPN model could not, suggesting that the ACP module could reject the detection of some tiny ships by considering them as back-ground noise, when using the shallow high-resolution information. Further investigation is still needed in the future to address this issue.

The attention mechanism is currently highly intertwined with transformers. Although the attention mechanisms used in the transformers and the multi-scale attention mechanism (MSAM) proposed in our manuscript all recalibrate the weights of the input features through global operations, so that features with stronger correlation can have more attention [26], the proposed MSAM is different in several aspects. First, most transformer-attention mechanisms used in the natural language processing (NLP) and computer vision fields, e.g., the multi-head attention (MHA) used by Vision Transformer [62], take sequential subblocks or substrings of the original data as the input, while MSAM uses the entire image as the input for feature extraction. Second, MHA is a self-attention mechanism while MSAM is a channel-attention mechanism. Typically, the self-attention mechanism is used as a spatial-attention mechanism to capture global information [63], and it is often employed to process single-scale feature information but some multi-scale feature information may be ignored. However, channel information is very different from spatial information because distinct channels in different feature maps usually represent distinct objects. MSAM is a channel-attention mechanism that adaptively recalibrates the weight of each channel, which can be viewed as an object selection process determining which channel should gain more attention. Third, MSAM has utilized multi-scale information due to the use of the hierarchical residual-like structure to extract the multi-scale feature maps as the input, but the transformer-attention mechanisms are mostly single-scale.

Moreover, we believe that the proposed MS-FPN model is also applicable to other types of remote sensing data such as the optical remote sensing datasets DOTA and NWPU-VHR10 [64,65], as well as the general computer vision datasets such as DeepScores and WiderPerson [66,67]. The images in these datasets contains objects such as ships, aircraft, and cars, which also present the multi-scale property. MS-FPN could be a valuable tool for the detection and segmentation of small objects in these datasets. In the field of computer vision, the simultaneous detection and segmentation of small targets is currently an active research field.

One important progress of MS-FPN in detecting and segmenting small objects is the use of both the shallow and deep features. In the computer vision field, some researchers have used the shallow features in their models [68,69]. Lim et al. has adopted the attention

mechanism to gain shallow features of each stage and needs to improve the utilization rate of multi-scale information at each stage [68]. Nie et al. pays attention to the multi-scale utilization of the original image after dimension reduction, and the utilization of other shallow layers in the backbone needs to be improved [69]. Different from the existing works, our method takes both shallow and deep information into account and further uses this information for segmentation. Our way of designing network modules, using different dilation rates that are embedded between the backbone and FPN of each stage to gain rich multi-scale feature information from different resolution, is unique for ship detection and instance segmentation in SAR images.

In the computer vision field, there are already relatively extensive studies on the extraction of multi-scale features, such as the relatively mature model ASPP and the recent backbone network P2T [70], and multi-scale image analysis is still a popular research direction. ASPP adopts a large dilation rate and global pooling, usually resulting in a variety of large receptive fields, which is very appropriate in the analysis of common visible light images containing many large targets. However, in SAR images, the majority of the data are small ships, and large receptive fields may introduce unexpected background noise. Our model has taken the multi-scale property of marine ships into consideration; thus, the proposed ACP method has better performance than ASPP. Moreover, here we have used an improved attention mechanism, which is multi-scale, different from the single-scale attention mechanism models such as SENet and CBAM, and the proposed MASM is more suitable for SAR image analysis.

In all, we have demonstrated that the proposed MS-FPN model is capable of efficiently detecting and segmenting multi-scale marine ships, especially the small ones, in high-resolution SAR images by using the shallow high-resolution feature maps and by designing a novel multi-scale attention mechanism. In the future, we will apply MS-FPN to other types of image data obtained in the field of optical remote sensing and computer vision.

**Author Contributions:** Conceptualization, Z.S., C.M. and Z.Z.; methodology, Z.S., C.M. and Z.Z.; software, Z.S.; validation, C.M. and Z.Z.; formal analysis, Z.S., C.M. and Z.Z.; investigation, Z.S.; resources, C.M.; data curation, C.M. and Z.Z.; writing—original draft preparation, Z.S.; writing—review and editing, J.C. and Z.Z.; visualization, Z.S.; supervision, S.C.; project administration, S.C.; funding acquisition, Z.Z. and S.C. All authors have read and agreed to the published version of the manuscript.

**Funding:** This research was funded by the National Natural Science Foundation of China, grant number 61831012 and 61401105, and by the State Key Laboratory of Applied Optics, grant number SKLAO2022001A14.

**Data Availability Statement:** The core code of the paper can be downloaded from GitHub via the link: https://github.com/s2120200252 accessed on 1 January 2022.

**Conflicts of Interest:** The authors declare no conflict of interest.

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
