# Peer review of "A Multi-Scale Feature Pyramid Network for Detection and Instance Segmentation of Marine Ships in SAR Images"

_remotesensing, doi:10.3390/rs14246312_

Round 1
Reviewer 1 Report
The paper develops a new network architecture based on a multi‐scale pyramid structure so that detection and instance segmentation of marine ships in SAR domain can be done at the same time. The pyramid structure consists of the Atrous Convolutional Pyramid module and the multi‐scale attention mechanism module. The Atrous Convolutional Pyramid extracts multi‐scale features for multi‐scale marine ships and the attention module selects important features from different scales. Experiments are provided to demonstrate that the proposed architecture is effective and leads to state-of-the-art performance over existing methods. The paper explores a practically significant problem and reads well. I have the following comments to be incorporated, for the next round of reviews:1. Attention mechanism currently is highly intertwined with transformers. Please add some description to clarify the difference between your attention module and those used in the architecture of transformers. 2. I was wondering what happens if instead of maxpooling, average pooling is used instead? 3. There are recent works based on deep learning for ship detection that need to be discussed: a. Rostami, M., Kolouri, S., Eaton, E. and Kim, K., 2019. Deep transfer learning for few-shot SAR image classification. Remote Sensing, 11(11), p.1374. b. Nie, X., Duan, M., Ding, H., Hu, B. and Wong, E.K., 2020. Attention mask R-CNN for ship detection and segmentation from remote sensing images. IEEE Access, 8, pp.9325-9334. c. Zhang, Y., Guo, L., Wang, Z., Yu, Y., Liu, X. and Xu, F., 2020. Intelligent ship detection in remote sensing images based on multi-layer convolutional feature fusion. Remote Sensing, 12(20), p.3316. d. Wang, Z., Zhou, Y., Wang, F., Wang, S. and Xu, Z., 2021. SDGH-Net: Ship detection in optical remote sensing images based on Gaussian heatmap regression. Remote Sensing, 13(3), p.499. e. Li, L., Zhou, Z., Wang, B., Miao, L., An, Z. and Xiao, X., 2021. Domain adaptive ship detection in optical remote sensing images. Remote Sensing, 13(16), p.3168. The above works can be discussed in the Introduction section to provide context. 4. Please repeat experiments several times and on your tables report both the average and the standard deviation to make the comparison more informative. 5. Is it possible to release your code on GitHub for reproducibility purposes?
Reviewer 2 Report
In this paper, the author designed a model called multi‐scale feature pyramid network (MS‐FPN), which can detect and segment ships in SAR image. The author adds the Atrous Convolutional Pyramid (ACP) and the multi‐scale attention mechanism (MSAM) into the classic model Feature Pyramid Network (FPN) to get a better effect. The former is to help output shallow and deep features, and the latter is to provide a better learning effect for multi-scale features. The author proved the validity of the model by testing on the public dataset HRSID. The logic and experiments of the paper are complete, and it is an excellent work. However, after reading it, I have several questions that may improve the paper:
1. The author has two core innovations in the paper, that is, adding the Atrous Convolution Pyramid(ACT) and Multi‐Scale Attention Mechanism(MSAM) to the model FPN. I have a few questions about the component ACT: Its main method is to get different receptive fields by changing the dilation rates of dilated convolutions, the values the author uses here are 1,2 and 4 respectively (by the way, One of the values of dilation rates in Figure 6 seems to be wrong). The benefits of this change are obvious and are described qualitatively in Figure 7, showing that it is effective on some objects. But such modifications appear to be relatively common in papers on ship detection, such as the reference below [1]-[5], where similar methods are used, except that they use different combinations of dilation rates values. So what's the difference between the ACT approach in the paper and the approach in these five references? In the author's model, will the combination of values of different dilation rates lead to worse or better quantitative results?
[1] Wu J, Pan Z, Lei B, et al. LR-TSDet: Towards Tiny Ship Detection in Low-Resolution Remote Sensing Images[J]. Remote Sensing, 2021, 13(19): 3890.
[2] Guo Y, Zhou L. MEA-Net: A Lightweight SAR Ship Detection Model for Imbalanced Datasets[J]. Remote Sensing, 2022, 14(18): 4438.
[3] Zhang T, Zhang X. HTC+ for SAR Ship Instance Segmentation[J]. Remote Sensing, 2022, 14(10): 2395.
[4] Ke X, Zhang X, Zhang T. GCBANet: A Global Context Boundary-Aware Network for SAR Ship Instance Segmentation[J]. Remote Sensing, 2022, 14(9): 2165.
[5] Yu W, Wang Z, Li J, et al. A Lightweight Network Based on One-Level Feature for Ship Detection in SAR Images[J]. Remote Sensing, 2022, 14(14): 3321.
2. The author tested the performance of his own method and some other methods on the public dataset HRSID in Table 1. A public dataset means that different methods of different papers can be compared. However, the results listed in Table 1 of this paper seem to be lower than the test results described in some other papers for the same public dataset HRSID. I will give several examples here, such as references [6]-[10]. If my guess is correct, it means that the author's model does not perform as well as some existing models.How does the author explain this.
[6] Ke X, Zhang X, Zhang T. GCBANet: A Global Context Boundary-Aware Network for SAR Ship Instance Segmentation[J]. Remote Sensing, 2022, 14(9): 2165.
[7] He B, Zhang Q, Tong M, et al. An Anchor-Free Method Based on Adaptive Feature Encoding and Gaussian-Guided Sampling Optimization for Ship Detection in SAR Imagery[J]. Remote Sensing, 2022, 14(7): 1738.
[8] Zhang T, Zhang X. HTC+ for SAR Ship Instance Segmentation[J]. Remote Sensing, 2022, 14(10): 2395.
[9] Zhang L, Wang H, Wang L, et al. Filtered Convolution for Synthetic Aperture Radar Images Ship Detection[J]. Remote Sensing, 2022, 14(20): 5257.
[10] Yu W, Wang Z, Li J, et al. A Lightweight Network Based on One-Level Feature for Ship Detection in SAR Images[J]. Remote Sensing, 2022, 14(14): 3321.
3. In Figure 13, the authors compared the results of detection and segmentation under the conditions of 'Only ACM', 'Only MSAM', and 'ACP&MSAM' by using ablation experiment (by the way, 'ACM' appears to be miswritten here, please correct it). However, it seems to me that the author's discussion of this qualitative result is not sufficient. For example, in some SAR Images, 'Only MSAM' seems to be better than 'ACP&MSAM' in some targets. For example, compared with the ground truth, in the second line of pictures (counting from top to bottom), 'Only MSAM' has a better detection result for several targets on the left. Should the author explain the reason for this? At least, the author should clarify this point in the discussion. On the other hand, as an observation of Table 2, it can be seen that when the two parts of "Only MSAM" and "ACP&MSAM" are compared, the improvement of the latter one does not seem to be obvious enough at least in terms of qualitative numerical results. Have the authors conducted tests on more datasets to prove the validity and robustness of this improvement?
Reviewer 3 Report
It is suggested to add analysis and discussion on missing and false detection.
Round 2
Reviewer 1 Report
The authors have addressed my concerns and improved the manuscript.